# Seaweed and Dendritic Growth in Unsaturated Fatty Acid Monolayers

**DOI:** 10.3390/membranes12070698

**Published:** 2022-07-08

**Authors:** Florian Gellert, Heiko Ahrens, Harm Wulff, Christiane A. Helm

**Affiliations:** Institute of Physics, University of Greifswald, Felix-Hausdorff-Straße 6, D-17489 Greifswald, Germany; gellertf@uni-greifswald.de (F.G.); heiko.ahrens.ellierode@web.de (H.A.); wulff@uni-greifswald.de (H.W.)

**Keywords:** lipid monolayer, fractals, Marangoni flow

## Abstract

The lateral movement in lipid membranes depends on their diffusion constant within the membrane. However, when the flux of the subphase is high, the convective flow beneath the membrane also influences lipid movement. Lipid monolayers of an unsaturated fatty acid at the water–air interface serve as model membranes. The formation of domains in the liquid/condensed coexistence region is investigated. The dimension of the domains is fractal, and they grow with a constant growth velocity. Increasing the compression speed of the monolayer induces a transition from seaweed growth to dendritic growth. Seaweed domains have broad tips and wide and variable side branch spacing. In contrast, dendritic domains have a higher fractal dimension, narrower tips, and small, well-defined side branch spacing. Additionally, the growth velocity is markedly larger for dendritic than seaweed growth. The domains’ growth velocity increases and the tip radius decreases with increasing supersaturation in the liquid/condensed coexistence region. Implications for membranes are discussed.

## 1. Introduction

The phase diagrams of two-dimensional (2D) Langmuir monolayers of amphiphilic molecules show many states of matter that are the 2D analogs of the three-dimensional (3D) solid, liquid, and gaseous states of matter [1,2,3,4]. Therefore, one could assume that the characteristic nonequilibrium growth morphologies of 3D materials, such as dendrites and finger instabilities, have their counterparts in 2D. Indeed, “liquid-condensed” (LC) domains growing in a supercooled “liquid-expanded” (LE) monolayer exhibit fingering instabilities very similar to those found in bulk materials [5,6,7,8,9,10].

However, the growth mechanisms are different: fingering morphologies of 3D materials rely on generating latent heat at the moving liquid/solid interface. The diffusion of excess heat away from the interface proceeds more efficiently with a modulated interface (the “Mullins-Sekerka” instability [11,12]). In monolayers, heat generation at the LC/LE interface does not matter because the monolayer floats on a large volume of water that acts as an isothermal reservoir. Nevertheless, domains of fatty acid and lipid monolayers with growth instabilities leading to fractals have been observed with fluorescence [5,6] and Brewster-angle microscopy [7,8]. The latter has the advantage that no dye or additional marker molecules are required.

The LE/LC coexistence region of a monolayer differs from the liquid/solid coexistence region of 3D materials in two ways: (i) the dependence of surface tension on lipid concentration, and (ii) the unusually large difference in area density between the LE and LC phases (50 to 200%). Therefore, to sustain the growth of an LC domain, efficient transport of lipid molecules from the LE to the LC phase must occur. Two transport mechanisms in the LE phase are distinguished: surface diffusion within the lipid monolayer or hydrodynamic flow of the subphase (Marangoni effect).

Bruinsma and coworkers proposed this idea first [13]; their theoretical calculations were based on comparing surface and adjective flow. They predicted two distinctly different growth instability classes: (i) seaweed domains that grow by surface diffusion in the LE phase or (ii) dendritic domains whose growth is determined by the adjective flow beneath the lipid monolayer. The agreement was qualitative, but a quantitative comparison failed [7,14]. According to theory, seaweed growth should not occur because the surface viscosity (the viscosity in the lipid monolayer) is a few orders of magnitude too high [14,15]. We cannot resolve this issue, but to better understand the different growth instabilities, we quantify the shapes of seaweed and dendritic domains as a function of (i) the compression speed of the monolayer and (ii) the supersaturation in the LE phase. 

The growth instabilities are determined by the drift velocity of the molecules in the LE phase towards the LE/LC domain boundary. Therefore, a meaningful parameter is the growth speed of the domains since it is proportional to the drift velocity of the molecules [13]. According to theory, the growth instabilities of the seaweed domains are less pronounced: their branches have tips with a larger radius. Furthermore, the separation between the branches is larger. This has consequences on the fractal dimension, which we determined.

In the past, dendritic growth and fingering instabilities were induced by pressure jumps, i.e., sudden increases in the supersaturation [6,7,9]. We decided to use a constant compression velocity because we wanted to have well-defined hydrodynamic conditions to investigate the time dependence of the domain parameters. Furthermore, domains that nucleated at different supersaturation levels within the coexistence region could be compared. We found that the compression speed determines if seaweed-like or dendritic domains grow.

As a model system, we have chosen an erucic acid monolayer [16]. The isotherms showed a low transition pressure with a broad coexistence region at the selected conditions (low temperature, low pH). The coexistence region was not flat, indicating a decreasing molecular area of the LE phase during compression. To quantify this supersaturation, the lateral pressure of the LE/LC phase transition $\pi_{\infty \;}$ at equilibrium was measured with low compression velocity $v_{c}$. We observed that domain nucleation occurred at lateral pressures slightly above $\pi_{\infty }$ and continued within the coexistence region. The excess lateral pressure $\Delta \pi =\pi_{\infty }-\pi$ was found to be a convenient parameter since it is proportional to supersaturation at low values of Δπ. The supersaturation concentration Δ*c* in units of ${Å}^{-2}$ was calculated from the area compressibility of the LE phase (cf. Appendix A).

The domains were imaged with Brewster Angle Microscopy at the beginning of the coexistence regime when isolated domains grew and the flow of the lipids in the LE phase was not influenced by neighboring domains. Their fractal dimension was calculated as outlined in Appendix B.

## 2. Materials and Methods

### 2.1. Materials

Erucic acid was purchased from Sigma-Aldrich (Merck KGaA), Darmstadt, Germany (purity ≥ 99%, according to supplier). To obtain the acidic subphase (pH 3), pure 37% muriatic acid (HCl) was used from Merck, Darmstadt, Germany. The pure water was provided by a Milli-Q Synthesis system with a nominal conductance of 0.054 μS.

### 2.2. Pockels-Langmuir Trough and Isotherms 

Compression surface isotherms (π−A isotherms) are recorded on a Teflon trough (Riegler & Kirstein, Potsdam, Germany). A Wilhelmy plate surface pressure sensor with a filter paper as a plate (accuracy of 0.1 mN/m) was used. The trough area is 3.5 × 30 cm^2^. The compression speed can be varied. The experiments were performed in ambient air. The trough temperature was kept constant ±0.1 °C with a thermostat (DC-30 Thermo-Haake, Haake Technik, Karlsruhe, Germany). The fatty acid was dissolved in chloroform solution (*c* = 0.1 mM). The solution was spread with a 100 μL syringe (model 1710, Hamilton, Bonaduz, Switzerland) and the chloroform was allowed to dissipate for a few minutes. Then, the monolayer was compressed with a predetermined compression speed and the isotherm was recorded.

### 2.3. Brewster Angle Microscopy (BAM)

The lipid films were studied by Brewster angle microscopy (BAM). A nanofilm_ultrabam from Accurion (Göttingen, Germany) was used to record real-time grayscale movies of the dendrite growth. Real-time grayscale videos were recorded at 20 frames per second with an image covering a surface area of about 0.24 mm^2^ (using Scheimpflug’s principle), corresponding to 1360 pixels × 1024 pixels and a spatial resolution of 2 µm. Due to the implemented Scheimpflug optics, it is possible to generate an overall focused image. However, the obtained images are distorted. The rectification and the background correction are performed by Accurion_Image (Accurion, Göttingen, Germany, version 1.2.3.).

### 2.4. Image Processing

#### 2.4.1. Contrast Enhancement

To examine different properties of the dendritic growth behavior, it is important to have sharp-edged structures. The edge of the domains is limited by the resolution of the camera and the contrast of the image. Therefore, the generated BAM images were contrast-enhanced with a combination of ImageJ (version 1.53i9) and Matlab (version R2021a). Further investigations, for instance, the determination of the fractal dimension, required a black and white image, which was created with a Matlab routine, separating every pixel with gray values above a predefined threshold as domain and pixels with gray values below as background, respectively. 

#### 2.4.2. Determination of the Fractal Dimension

The fractal dimension of the structures was determined with a boxplot algorithm, developed by F. Moisy [17]. A detailed description of the calculation can be found in Appendix B.

## 3. Results and Discussion

### 3.1. Isotherms of Erucic Acid Monolayers at Different Compression Velocities

We studied acid molecules with uncharged head groups. Therefore, isotherms of erucic acid were recorded at low subphase pH (pH = 3). The temperature was kept constant at T=10 °C. To approach equilibrium thermodynamics, the monolayer was compressed with a low compression velocity of $v_{C}=1.2\;{Å}^{2}/\left(\mathrm{molecule}\cdot \min\right)$. The onset of the lateral pressure increase occurs at a molecular area of $48\;{Å}^{2}$. At further compression, the lateral pressure increases slowly and monotonously until a kink occurs, which marks the phase transition pressure $\pi_{t}\overset{!}{=}\pi_{\infty }$=12.4 mN/m. The corresponding molecular area is $A_{t}\overset{!}{=}A_{\infty }=27.5\;{Å}^{2}$. On further compression, the so-called coexistence regime is reached. While the molecular area decreases, the lipids undergo a phase transition from the LE to the LC phase. In the LC phase, the alkyl chains are ordered [2,3]. In the coexistence regime, the increase in lateral pressure is smaller than in the LE phase [16,18]. Compared to phospholipid monolayers, the pressure increase in the coexistence regime is rather steep [3,19,20]. Once the molecules are ordered, further compression in the LC phase leads to a steep pressure increase. Eventually, at a lateral pressure of 25 mN/m, the molecular area is 20 Å^2^ [16].

The influence of the compression speeds $v_{C}$ on the isotherms has been investigated. Figure 1 shows three typical isotherms of different compression velocities. The blue curve represents a slowly compressed monolayer close to the thermodynamic equilibrium. The other ones (black and red) are isotherms measured at higher compression velocities, which did not allow relaxation towards the thermodynamic equilibrium. 

At large molecular areas, the isotherms are very similar. With the increase in the compression speed, the LE/LC phase transition occurs later, i.e., the lateral pressure $\pi_{t}$ is increased while the molecular area $A_{t}$ is decreased. From the inset of Figure 1, the intervals $A_{\infty }-A_{t}$ and $\pi_{t}-\pi_{\infty }$ are plotted for different compression speeds $v_{C}$. The slope of the line in the inset shows an increase in excess lateral pressure per molecular area decrease of $-1.38\;\mathrm{mN}/\left(m\cdot {Å}^{2}\right)$.

### 3.2. Domain Growth Visualized with Brewster Angle Microscopy (BAM) Videos 

The growth of domains was observed with Brewster Angle Microscopy (BAM). Domains started to grow at the beginning of the coexistence region, with a slight delay. Videos and isotherms were recorded simultaneously.

Figure 2 shows typical examples observed with different compression velocities $v_{C}$. Depicted is a time series of contrast-enhanced BAM images. At a low compression velocity (left, $v_{C}$ = 1.2 ${Å}^{2}/\left(\mathrm{molecule}\cdot \min\right)$), a few domains nucleate. Figure 2 shows the growth of one domain during 72 s. These domains grew to a diameter of several 100 μm. The shape of the domains is somewhat arbitrary, and their side arms have a significant and not very well-defined separation. The described features are typically for seaweed domains [7].

At a high compression velocity (Figure 2, right column, $v_{C}$ = 2.3 ${Å}^{2}/\left(\mathrm{molecule}\cdot \min\right)$), a significantly higher number of domains nucleate and grow, in agreement with literature [21]. On further monolayer compression, the domains start interacting with each other, limiting their final size. After 62 s, the domain growth led to a carpet-like, wholly covered area. Compared to the seaweed domains, this suggests a faster domain growth. In contrast to the seaweed domains, the structures distinguish themselves by relatively thin and more straightened main and side branches. Additionally conspicuous is the significantly higher number of side branches, leading to needle-like forms. The described features are typical for dendritic growth [7,13]. We conclude that depending on the compression velocity $v_{C}$, either seaweed or dendritic domains grow.

### 3.3. Parameters Characterizing Domain Growth

#### 3.3.1. Influence of the Compression Velocity $v_{C}$ on Fractal Dimension $D_{F}$

Fractals are visible in the observed structures. The fractal dimension $D_{F}$ has been determined. The complexity of a pattern is quantified by the ratio of the change in detail to the change in scale [22]. A detailed description of the procedure is given in Appendix B. We determined the fractal dimension of a complete image, not only of a single domain. The limitation was the resolving power of the BAM. Figure 3 shows the fractal dimension’s evolution as a function of time. At the beginning (t < 5 s), the domains are small, and so is the fractal dimension. The diameter of the domains is similar to the resolution of the BAM (2 μm). Therefore, the fractal dimension at an early growth stage has a broad error. The increase in the fractal dimension at the beginning is attributed to adding branches to the domain nuclei. After about twenty seconds, the fractal dimension is constant (within error). Additionally, Figure 3 shows that at a low compression velocity $v_{C}$, the fractal dimension $D_{F}$ of the seaweed domains is considerably smaller than for dendrites grown at a larger $v_{C}$. For seaweed domains, $D_{F}=1.61$ has been measured, whereas $D_{F}$ is above 1.85 for dendritic domains. In the experiments shown, the fractal dimension $D_{F}$ increases with $v_{C}$. The larger number of side branches leads to more complex structures and causes an additional increase in $D_{F}$. Concluding, the fractal dimension is an indicator to distinguish between the two growth classes with different pattern evolution.

#### 3.3.2. The Growth Speed $v_{R}$ of the Domains

In Figure 2, the different time scales for LC domains in seaweed and dendritic growth regimes were apparent. To quantify this observation, the growth speed $v_{R}$ was determined. Figure 1 shows the investigated properties of a domain. To determine the growth speed $v_{R}$ we focus on the main branches (cf. Figure 4, bottom). The branch length l defines the length of a main branch. From its time-dependent increase, the growth speed $v_{R}$ is calculated (cf Figure 4, bottom). 

A typical video of this study contains the images during the complete compression of the erucic monolayer at a defined compression velocity. We focus on the part of the movie which shows the nucleation until the domains reach a very high surface coverage, when the different side branches can no longer be resolved. After a contrast-enhancement procedure, the boundaries of the domains were analyzed in detail. To determine the growth speed, a selected domain was compared in the different frames of a movie. The formalism of the determination of $v_{C}$ is illustrated in Figure 4 (bottom). The length of a main branch was measured with the software ImageJ, then the length was converted from pixels into micrometers. Every 50 ms the camera recorded one image. The growth speed $v_{C}$ is the quotient of the increment in branch length, Δl per time increment, delta Δt. Domains often drifted out of the image section of the CCD camera, which limited the observation time. Furthermore, we focused on isolated domains to avoid domain distortion by adjacent domains. 

The growth kinetics of the domains are further analyzed in Figure 4 (top). Three different monolayers with three different compression velocities $v_{C}$ were analyzed. The slowest compression speed leads to seaweed domains, the other ones to dendrites. Plotted is the time-dependent main branch length l. The main branches of dendrites showed the same growth speed. Therefore, each domain is represented by one main branch. One exception is the two upper blue lines (2.1 μm/s, 3.0 μm/s) which represent two branches of the same seaweed domain, growing in different directions. The different growth speed is attributed to the somewhat arbitrary structure of seaweed domains (cf. Figure 2). The length of each investigated main branch increases linearly with time. Lines in Figure 4 (top) are least-square fits; the slope corresponds to the growth speed $v_{R}$, indicated by the numbers beside the respective lines. Therefore, each main branch exhibits a constant growth speed $v_{R}$. However, the domains grown at a higher compression velocity $v_{C}$ show a faster growth speed $v_{R}$. If one monolayer is considered, the main branches of domains that nucleated at later times grow faster (cf. Figure 4, top). Note that later times indicate lower molecular areas and higher excess lateral pressure. Depending on the film parameters, the growth speed $v_{R}$ varies by an order of magnitude, from 2.1 µm/s to 24.8 μm/s.

#### 3.3.3. Dependence of Growth Speed $v_{R}$ on Compression Velocity and Supersaturation

In Figure 5, the dependence of the growth speed $v_{R}$ on the excess lateral pressure $\Delta \pi =\pi -\pi_{\infty }$ is quantified. There are two contributions to the excess lateral pressure Δπ, the increase in the phase transition pressure $\pi_{t}$ (cf. Figure 1) due to the compression velocity and the additional increase due to the non-flat coexistence regime. At the lowest compression velocity $v_{R}$, the growth of seaweed domains starts at low excess lateral pressures (Δπ≈0.5 nm), with the lowest growth velocity $v_{C}$ observed (2.1 µm/s). The growth speed of domains that nucleate later in the coexistence regime is about a factor of two larger. Monolayers that were subject to a larger compression velocity $v_{C}$ show dendritic growth. The dendrites start to grow at larger excess lateral pressures Δπ than seaweed domains. The influence of the different domain growth kinetics on the growth speed $v_{R}$ can be best seen at the excess lateral pressure Δπ≈2.0−2.4 mN/m. At this excess lateral pressure, the main branch of a seaweed domain grows at 4.9 μm/s, while the dendrite one grows twice as fast at 10.2 μm/s (cf. Figure 5) (respective compression velocities are 1.2 and 2.3 ${Å}^{2}/\left(\mathrm{molecule}\cdot \min\right)$).

The largest variation in growth speeds within one monolayer is observed for the monolayer which was compressed with $v_{C}=2.3\;{Å}^{2}/\left(\mathrm{molecule}\cdot \min\right)$ and exhibited dendrites. At the beginning of the coexistence regime, at low excess lateral pressure (1 mN/m), the growth speed is 6.0 μm/s. At the end of the coexistence regime, the excess lateral pressure has quadrupled (3.8 mN/m), and so has the growth speed (24.8 μm/s). A slightly larger compression velocity ($2.5{\;Å}^{2}/\left(\mathrm{molecule}\cdot \min\right)$) leads to delayed nucleation at an excess pressure of 4.2 mN/m and a large growth speed, which varies little during further compression (between 11.3 and 16.9 µm/s). 

In Appendix A, the supersaturation $\Delta c=1/A-1/A_{\infty }$ is calculated in dependence on the excess lateral pressure $\Delta \pi =\pi -\pi_{\infty }$. A linear relationship could be derived for small excess lateral pressures, which are found in the coexistence regime of erucic acid (cf. Figure 1). A shift in Δπ of 1 mN/m corresponds to a change in supersaturation of around $1.1\times 10^{3}\;{Å}^{-2}$; or a relative change in the supersaturation of 2.6%. This small change can have a pronounced effect on the growth speed if compression velocity and additional supersaturation in the coexistence are suitable. This observation is in agreement with theoretical predictions [13].

To summarize, seaweed domains occurring at a lower monolayer compression velocity nucleate at a lower excess lateral pressure than dendrites; their main branches have a smaller growth speed. Once the main branch of a domain starts to grow, its growth speed is constant. The growth kinetics are established when the branch is nucleated. The constant growth speed is independent of the domain’s shape, fractal dimension, or growth class. Furthermore, for any monolayer, a shift toward higher growth speed occurs for branches nucleating at higher initial excess lateral pressures.

#### 3.3.4. The Influence of Excess Lateral Pressure Δπ and Supersaturation Δc on the Tip Radius r

Tip radii *r* of the main branches are sketched in Figure 1. To measure them, the video images have been contrast-enhanced, as described before. Once formed, the tip radius of a domain does not change, while the domain grows (until the domain leaves the field of view). Figure 6 shows the dependence of the tip radii on the lateral excess pressure for the three investigated monolayers with different compression velocities $v_{C}$. Always, the tip radii decrease with increasing excess lateral pressure (supersaturation, respectively). The highest reduction was found for the seaweed domains: the tip radius decreases from nearly 3 μm to 1.3 μm, while the lateral pressure increases by 2.2 mN/m. The tip radii of dendrites are smaller: they start at 1.2 µm and decrease to 0.95 µm. 

For all structures, a linear decrease in the tip radii with increasing supersaturation was observed, in agreement with theoretical predictions [23,24]. In the past, the large tip radii of seaweed domains in lipid monolayers were taken as a hallmark of the seaweed domains [7]. We find that this correlation has to be used with some care, since the tip radius depends, for seaweed domains, sensitively on the lateral pressure. For dendrites, the dependence of the tip radius on the lateral pressure is weaker.

#### 3.3.5. Side Branch Separation λ for Seaweed Domains and Dendrites

The side-branch separation is sketched in Figure 1. Theoretically, side branches of dendrites should be closer to each other than of seaweed domains [13]. A comparison of the BAM pictures seen in Figure 2 for seaweed-domains and dendrites suggests that the side branches of the seaweed domains appear more irregular, and their formation is more arbitrary. Furthermore, they are farther apart. To quantify this behavior, the side branch separation λ was measured for monolayers compressed at different velocities $v_{C}$. Figure 7 shows the findings, averaged over 20 measurements from different domains. The separation of the side branches is independent of the excess lateral pressure. It numerically confirms the visual observations. Indeed *λ* is significantly larger in the seaweed than in the dendritic growth regime. The large error bars found in the seaweed regime indicate the higher irregularity of the domain shape. This leads to the conclusion that the driving mechanism of irregular growth, propagating with a mode q=2 π/λ at the LE/LC boundary of the growing domain, is different in the two growth regimes, in agreement with theoretical predictions [11,13,24].

#### 3.3.6. Influence of the Compression Velocity on the Flow in the LE Phase 

Up to now, mesoscopic quantities of the domains have been analyzed, such as main branch growth velocity, the average separation between side branches, and the tip radius. With these parameters, the growth kinetics of seaweed domains could be characterized. The domain growth is made possible by the flow in the LE phase towards the domains. We cannot measure the flow next to the domains directly, but we can estimate the flow velocity far away from a domain, $v_{\infty }$, but flowing toward the domain. We estimated $v_{\infty }$ from the main branch growth speed $v_{R}$. The latter is influenced by the compression velocity $v_{C}$ that is calculated from the velocity of the barrier of the Langmuir trough, $v_{Barrier}$, and the dimensions of the Langmuir trough. 

A measurement was carried out in the coordinate system of the laboratory, hence the LE/LC boundary moves. Since the number of amphiphilic molecules is constant, one can state [13]:(1)$$v_{\infty }\cdot c_{\infty }=v_{R}\cdot (c_{S}-c_{0})$$
with $c_{\infty }$ the lateral concentration in the LE phase far away from the boundary of the domains. $c_{S}$ denotes the surface concentration in the LC phase and $c_{0}$ the surface concentration in the LE phase close to the boundary domain.

For erucic acid, $c_{\infty }$ has been calculated from the molecular area at the LE/LC phase transition (1/27.5 ${Å}^{2}$), $c_{S}$ from the molecular area determined by X-ray diffraction [16] in the LC phase (1/20 ${Å}^{2}$). The surface concentration of the LE phase $c_{\infty }$ is reduced close to the domain by a location-dependent excess surface concentration Δc’ to $c_{0}$ [25],
(2)$$c_{0}=c_{\infty }-\Delta c’$$

Rearranging Equation (2) and assuming that $c_{0}$ does not deviate much from $c_{\infty }$ (true within 10%, cf. Table A1 in Appendix A), a dependency of the flow speed $v_{\infty }$ from the growth speed $v_{R}$ can be found:(3)$$v_{R}=\frac{c_{\infty }}{c_{S}-c_{0}}v_{\infty }\approx \;\frac{c_{\infty }}{c_{S}-c_{\infty }}v_{\infty }$$

Using the numbers from the erucic acid monolayers (cf. Figure 1), one obtains
(4)$$v_{R}=\frac{\frac{N}{27.5}}{\frac{N}{20}-\frac{N}{27.5}}v_{\infty }\approx 2.67v_{\infty }$$

N denotes the number of molecules in the monolayer. Note that $v_{\infty }$ is about a factor of three slower than the growth speed of the main branch of a domain. The values for the flow velocity are listed in Table 1. $v_{\infty }$ calculated from the different growth speeds is shown in Table 1, which contains representative values of $v_{R}$ deduced from Figure 4 (top).

Note that the velocity of the barrier is larger than the growth speed of the main branch. This is to be expected because the barrier speed moves in one dimension, however, the domains grow in two dimensions. Table 1 allows us to qualitatively compare $v_{R}$ with $v_{Barrier}$:(5a)$$v_{Barrier}\approx \;24.37\cdot v_{R,Seaweed}$$
(5b)$$v_{Barrier}\approx \;12.18\cdot v_{R,Dendrite}$$

This result is consistent with the experimental observation that the growth speed of domains is higher than the barrier velocity. It also suggests that dendrites grow generally faster than seaweed domains.
(6)$$\frac{v_{R,\;\;Dendrite}}{v_{R,Seaweed}}\approx 2$$

This shows that the hydrodynamic flow of the subphase causes a larger growth velocity.

## 4. Conclusions

We used uncharged monolayers of erucic acid to describe the different growth instability classes. Theoretically, seaweed growth is predicted when lipid diffusion dominates, whereas dendritic growth is expected when adjunctive diffusion contributes to the lipid movement [13]. The monolayer was especially suitable for these studies because in the LE/LC coexistence region, the lateral pressure and, thus, the supersaturation increased. By varying the compression speed, either seaweed or dendritic growth was obtained. 

Using Brewster Angle Microscopy (BAM, Accurion, Göttingen Germany), we analyzed the shape of the domains. The fractal dimension of seaweed domains was lower than that of dendritic domains, a feature described for a few other lipid monolayers [7]. The main branches of seaweed domains have a smaller growth speed than dendrites and the separation of the side branches is larger and shows more scatter. The tip of the main branch has a larger radius.

We compared the domains of monolayers compressed with the same compression velocity, but which nucleated at different degrees of supersaturation within the LE/LC coexistence regime. With increasing supersaturation (excess lateral pressure), the radii of the tips of the main branches decreased while their growth speed increased. The former feature has been predicted theoretically [23,24], the latter is new (to the best of our knowledge). In addition, the main branches of dendrites have a growth speed of about a factor of two greater than the main branches of seaweed domains. The faster growth speed is seen as evidence of adjunctive flow. 

Finally, we would like to compare the unidirectional compression velocity of the monolayer (75 µm/s to 200 µm/s) with the speed of blood (between 500 µm/s and $2.5\times 10^{5}\;µm/s$). These numbers suggest that the subphase does influence lipid movement. We find that the detailed study of domain growth in lipid monolayers is a tool to explore the different flow mechanisms which cause lipid movement in biological membranes.

## Figures and Tables

**Table 1 membranes-12-00698-t001:** Comparison of velocities influencing domain growth. $v_{C}$ is the compression velocity calculated from the dimensions of the Langmuir trough and the barrier velocity $v_{Barrier}$. $v_{R}$ is the growth velocity of the main branch and $v_{\infty }$ the flow velocity in the LE phase of molecules far away from the domain, yet flowing already toward the domain.

$v_{C}$ $\left[{Å}^{2}/\left(\mathrm{molecule}\cdot \min\right)\right]$	$v_{Barrier}$ [cm/min]	$v_{Barrier}$ [μm/s]	$v_{R}$ [μm/s]	$v_{\infty }$ [μm/s]
1.2	0.46	76.67	2.1–4.9	0.8–1.2
2.3	1.08	180.00	6.0–24.8	3.8–5.5
2.5	1.20	200.00	11.3–16.9	4.3–6.3

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
