# Peer review of "Seaweed and Dendritic Growth in Unsaturated Fatty Acid Monolayers"

_membranes, 2022, doi:10.3390/membranes12070698_

Round 1
Reviewer 1 Report
In this manuscript, the formation of pattern domains in the liquid/condensed coexistence is investigated, with many exciting measurements. They determined the dimension of the fractal domains and showed that by increasing the monolayer compression speed, a transition from seaweed growth to dendritic growth is induced. Also, they found that seaweed domains form broad tips and wide and variable side branch spacing, and dendritic domains have a higher fractal dimension, narrower tips, and small, well-defined side branch spacing. The growth velocity is markedly larger for dendritic than seaweed growth. The domains' growth velocity increases, and the tip radius decreases with increasing supersaturation in the liquid/condensed coexistence region. Measurements will give quantitative information for future theoretical developments. I like the information given in the paper. I think that the paper has to be accepted for publication.
Just one minor thing:
At the beginning of page 6,
……At low compression velocity (left, ??=2.3 â„« /(molecule ∙ min)), few domains nucleate………… Is the concentration right?
Reviewer 2 Report
I recommend publishing this manuscript due to the interesting topic of the discussed model membrane and the scientific level of the presented experiments. Indeed, the investigated parameters of the condensed domains formed in the LE-LC phase transition and the interrelationship between them may play an important role in various human physiological processes. The results of the experimental studies presented in this manuscript provide in-depth insight into the kinetics of the domain growth, the analysis of the shape and size of the domains, and the relationship between the compression rate of the lipid monolayer and the flow in the LE phase. These results may be useful to other researchers.
Minor changes:
- The purity of the erucic acid used in the experiments must be reported.
- Page 6, line 180, should be: vc = 1.2 Å2/molecule·min
- Page 12, line 377: is “TABLE 1A” should be “TABLE A1”
- Page 13, line 382: The definition of N is not correct, it is for molecules in the monolayer, not in the trough.
- Page 15, Table A1, 1st row: the visibility of the units should be improved
- In several places in the manuscript (e.g. lines 58, 146, 186, 364, 373-374) I noticed problems with citing the references. The names or data of the journal appear instead of the numbers.
The manuscript is well written. All conclusions are supported by data. The figures are carefully prepared allowing the reader to have a clear picture of all aspects discussed. Summing up, I believe that after minor corrections, the work should be published in Membranes.
